# Enhancing Image-Guided Radiation Therapy for Pancreatic Cancer: Utilizing Aligned Peak Response Beamforming in Flexible Array Transducers

**DOI:** 10.3390/cancers16071244

**Published:** 2024-03-22

**Authors:** Ziwei Feng, Edward Sun, Debarghya China, Xinyue Huang, Hamed Hooshangnejad, Eduardo A. Gonzalez, Muyinatu A. Lediju Bell, Kai Ding

**Affiliations:** 1Department of Radiation Oncology and Molecular Radiation Sciences, Johns Hopkins University School of Medicine, Baltimore, MD 21287, USA; zfeng15@jhmi.edu (Z.F.); edwardsun12895@g.ucla.edu (E.S.); hamed@jhu.edu (H.H.); 2Department of Electrical and Computer Engineering, Johns Hopkins University, Baltimore, MD 21218, USA; egonza31@alumni.jh.edu (E.A.G.); mledijubell@jhu.edu (M.A.L.B.); 3Department of Computer Science, University of California, Los Angeles, CA 90095, USA; 4Department of Biomedical Engineering, Johns Hopkins University School of Medicine, Baltimore, MD 21287, USA; dchina1@jhmi.edu (D.C.); xinyue@gatech.edu (X.H.); 5Department of Computer Science, Johns Hopkins University, Baltimore, MD 21218, USA

**Keywords:** ultrasound image, image guidance radiotherapy, ultrasound beamforming

## Abstract

**Simple Summary:**

In this study, we focus on improving radiation therapy (RT), particularly advances in managing the intra-refractive movement of pancreatic tumors and surrounding tissues during treatment. It is hard to track internal anatomy changes, such as those induced by respiration, in conventional RT, which can lead to the inadequate treatment of pancreatic cancer as well as cause potential harm to surrounding normal tissues. To address this issue, we focused on the use of ultrasound imaging, specifically the use of novelty flexible array transducers, for the real-time monitoring of these movements. However, challenges arise due to the nature of flexible array transducers that change with the shape of the body. Our study developed a new method, the Aligned Peak Response (APR) method, and combined it with an auxiliary structure with embedded markers. The method aims to improve the accuracy of the beamforming and motion tracking of ultrasound images during RT. We tested the effectiveness of the method using simulation and in vitro data aimed at improving RT accuracy and reducing patient risk.

**Abstract:**

To develop ultrasound-guided radiotherapy, we proposed an assistant structure with embedded markers along with a novel alternative method, the Aligned Peak Response (APR) method, to alter the conventional delay-and-sum (DAS) beamformer for reconstructing ultrasound images obtained from a flexible array. We simulated imaging targets in Field-II using point target phantoms with point targets at different locations. In the experimental phantom ultrasound images, image RF data were acquired with a flexible transducer with in-house assistant structures embedded with needle targets for testing the accuracy of the APR method. The lateral full width at half maximum (FWHM) values of the objective point target (OPT) in ground truth ultrasound images, APR-delayed ultrasound images with a flat shape, and images acquired with curved transducer radii of 500 mm and 700 mm were 3.96 mm, 4.95 mm, 4.96 mm, and 4.95 mm. The corresponding axial FWHM values were 1.52 mm, 4.08 mm, 5.84 mm, and 5.92 mm, respectively. These results demonstrate that the proposed assistant structure and the APR method have the potential to construct accurate delay curves without external shape sensing, thereby enabling a flexible ultrasound array for tracking pancreatic tumor targets in real time for radiotherapy.

## 1. Introduction

Radiation therapy (RT) is one of the three conventional treatments for cancer, notable for its ability to eliminate tumors while sparing the surrounding healthy tissues and organs [1]. The primary challenge of radiation therapy, especially stereotactic body RT (SBRT), lies in delivering a high spatial dose accurately to the tumor target amid significant changes in the internal anatomy during the treatment process [2]. The use of image-guided radiotherapy (IGRT) before treatment aims to mitigate inter-fraction motion, such as patient setup errors and day-to-day anatomical variations due to weight loss, tumor progression, or tumor shrinkage. However, intrafraction motion—caused by breathing, tumor baseline shifts or drifts, and peristalsis during the treatment—requires an increase in the internal target volume and the internal margin to account for such motions, which in turn raises the risk of collateral damage to the normal tissues surrounding the target. The need to track and monitor intrafraction motion becomes more critical when considering factors like faster anatomical motions and larger drifts caused by respiration or cardiac activity. Previous research has shown that the respiration process can lead to several centimeters of target translation, undesired rotation, and significant deformation in the treatment of liver [3,4,5,6,7], lung [8,9,10,11,12,13], and pancreatic cancers [14,15,16]. Cardiac activity has also been shown to similarly affect the position of mediastinal lymph nodes [8,11,17,18], and pancreatic or liver tumors [3,19].

Intrafraction motions of the tumor and surrounding normal tissues can lead to the tumor receiving an insufficient dose while exposing organs at risk (OARs) to overdoses if a treatment based on static anatomy is applied [2,20]. One strategy for protecting normal tissue involves minimizing the internal target volume and internal margin (ITV) through beam gating, which relies on real-time image tracking and monitoring [2]. This approach entails expanding the gross tumor volume to ITV with a margin, thereby designing the planning target volume (PTV) during the planning process to counteract the adverse effects of intrafraction motions during treatment and ensure target coverage [20,21,22]. However, the downside of this margin method is an increased risk of concurrent damage to OARs [20,23,24]. Even though gating techniques [25], which aim to reduce such margins while maintaining target coverage, necessitate a real-time movement tracking technique to manage the beam’s on/off signal during treatment, they represent a critical step forward in addressing this complex issue.

Ultrasound imaging, known for being real-time, portable, non-ionizing, and cost-effective, has found extensive use in the medical field [26,27]. Recently, the feasibility and efficiency of implementing ultrasound imaging for tracking abdominal target movement during radiotherapy have been tested and evaluated [28,29,30,31,32]. However, the widespread clinical adoption of conventional ultrasound transducers for radiotherapy guidance is limited by two primary factors. First, the use of conventional ultrasound transducers requires professional training and substantial clinical experience [33]. Second, the rigid casing of the conventional transducer necessitates placement on the patient’s body using external technologies, such as passive robotic arms, which can introduce setup errors due to contact pressure. This pressure may cause variations in intra- and inter-fraction anatomical movements, potentially affecting treatment accuracy [27,28,30,34].

The flexible probe emerges as a novel solution, aiming to overcome the limitations and disadvantages associated with the conventional rigid transducer and to reduce uncertainties during treatments [27,35,36]. This innovative flexible array transducer is deformable, allowing it to easily conform to the arbitrary shapes of patients’ bodies. However, its deformability poses a challenge, as the geometry of the wearable flexible array transducer is generally unknown. This uncertainty arises from varying body surface shapes and changes induced by patient respiration. Consequently, traditional ultrasound beamformers, such as the delay-and-sum (DAS) method, which reconstruct B-mode images from radiofrequency (RF) channel data, may incorrectly calculate the time of flight (ToF) between each element and its focal point. This misalignment can result in inappropriate time delays for each scanline [37], highlighting the urgent need for a novel ultrasound beamformer to address this issue.

On one hand, external assistant tracking sensors represent a solution for monitoring the geometry changes in flexible array transducers. Previous research explored the feasibility of attaching three pairs of strain gauges to a transducer’s front and back surfaces. However, this approach restricts the transducer to forty-eight elements, resulting in a limited field of view (FOV) [38]. On the other hand, mathematical models and algorithms have been proposed to estimate the flexible transducer geometry and optimize shape parameters, aiming to enhance image quality [36,39,40,41]. The primary drawback of this method is the time-consuming nature of the estimation procedures.

Recent advancements and breakthroughs in deep learning have led researchers to propose an end-to-end deep learning approach for directly reconstructing high-quality B-mode images from RF channel data, thereby bypassing the delay estimation step [27]. In a prior study, three different neural networks were employed as a deep neural network (DNN) approach for estimating precise time delays for each scanline, summing delayed data, and processing post-image data. This DNN approach resulted in a reduction in the average full width at half maximum (FWHM) of point scatter targets by 1.80 mm in simulations and 1.31 mm in scan results. Furthermore, the contrast-to-noise ratio (CNR) of anechoic cysts improved by 0.79 dB in simulations and 1.69 dB in phantom scans with this method [27]. Additionally, another DNN-based method was proposed to estimate the geometry of the flexible transducer shape from RF data [42]. Simulation experiments demonstrated an average element position mean absolute error (MAE) of 0.86 mm, and average reconstructed image peak signal-to-noise ratio (PSNR) and mean structural similarity (MSSIM) of 20.6 and 0.791, respectively. In vivo experiments yielded an average element position MAE of 1.11 mm, with an average reconstructed image PSNR and MSSIM of 19.4 and 0.798, respectively [42]. These outcomes suggest the viability of this method for real-time monitoring with flexible transducer probes [42]. Nonetheless, the efficacy of this approach heavily relies on the quality and consistency of the training and testing data.

In this study, we propose a potential assistant structure with embedded markers and introduce an innovative alternative method, the Aligned Peak Response (APR) method, to modify the conventional delay-and-sum (DAS) beamformer for reconstructing ultrasound images from a flexible array transducer. Our proposed method leverages the geometry information of the flexible array transducer for ultrasound image beamforming and for tracking pancreatic tumor movement during radiotherapy. We assessed the feasibility and efficiency of the APR method and assistant structure using Field-II simulation data and experimental ultrasound RF data obtained from a tissue-mimicking phantom.

## 2. Materials and Methods

### 2.1. Methodology of APR

As shown in Figure 1A, a potential clinical application of our flexible probe was set up by placing an assistant structure embedded with markers (the red circle is an example of an assistant structure) on the patients’ body. The blue, green, and yellow rectangles represent three example elements. The peak responses generated by this marker are used for the APR method. Three example responses, named the first, reference, and third scanlines, are shown in Figure 1B. The blue, green, and yellow curves represent the strong reflections generated by the mark. The received time points of the peak responses in these three scanlines are t_1_, t_2_, and t_3_, respectively. In calculating the delays between peak response-received time points, d_1_ and d_2_ are calculated as follows (1) and (2):d_1_ = (t_2_ − t_1_)fs(1)
d_2_ = (t_3_ − t_2_)fs(2)

The d represents the delay sample numbers of each scanline, which results from the received time and sample rate. The units of t, fs, and d are s, Hz, and s, respectively.

**Figure 1 cancers-16-01244-f001:**
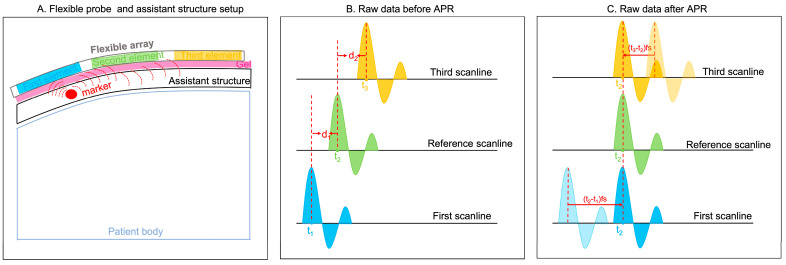
Principle of APR. (**A**) Flexible probe and assistant structure setup. Blue, green, and yellow rectangular are three example elements. The pink rectangular shade represents gel. The red circle represents the marker embodied within the assistant structure. (**B**) Raw data before APR. Three examples of peak response are shown in blue, green, and yellow curves. The peak responses of these three scanlines are received in t_1_, t_2_, and t_3_, respectively. (**C**) In aligning the peak response, the first and third scanlines are delayed in the d_1_ time samples and d_2_ time samples, respectively.

Therefore, for aligning peak responses, the first and third scanline responses are delayed the d_1_ time samples and d_2_ time samples, respectively.

### 2.2. Point Target Phantom Simulation Based on Field-II

Field II is a MATLAB-based simulation software package that is widely used to simulate realistic ultrasound images [43,44,45,46]. It models the physics of wave propagation, scattering, and transducer properties, allowing users to accurately simulate the performances of various ultrasound transducers and imaging scenarios by controlling different parameters, such as excitation frequency, the focus depth, the element positions and shapes, and the steering beam angles [47,48].

The Field II ultrasound simulation software was used to simulate a point or scatter phantom to mimic the assistant structure with embedded markers. This model was correspondingly used to generate RF data for testing the feasibility of our proposed APR method. We simulated two different types of imaging phantoms: a point target phantom that contained 1 or 2 point targets at different locations, and a scatter phantom with 10,000 scatters that contained a single anechoic or hypoechoic target in background tissue for testing the optimal assistant structure designed. The descriptions and results of the scatter phantom simulations are shown in the Appendix A.

In the point target phantom simulations, we simulated three point phantoms to mimic the assistant structure. The first point phantom included one point target in the center of the phantom. The lateral position (x) of this point target was 0 mm, and the axial position (z) was 30 mm. The second point phantom included two point targets and the lateral positions (x) and the axial positions (z) of these point targets were [−30 mm, 30 mm] and [30 mm, 30 mm], respectively. The third point phantom included one point target on the right side of the phantom, which the lateral position (x), and the axial position (z) of this point target was [30 mm, 30 mm].

The simulated transducer was modeled after a flexible array transducer (made by Hitachi and Japan Probe, Tokyo, Japan), where the sound speed was set to 1540 m/s. The simulated transducer in the point target phantom simulations was set as one active element in the emit aperture and one hundred twenty-eight active elements in the receive aperture.

The overall workflow of the Field-II simulation is illustrated in Figure 2. First, before applying the APR method, data normalization and smoothing calculations were included in our pre-processing workflow for highlighting the peak response and minimizing the effect caused by irrelevant signal (signals are not reflected by the markers) in the RF channel data obtained from all simulated phantoms. The normalization was performed in the simulated RF data by dividing the data by the maximum value of the entire RF channel dataset. Then, the 1st round-estimated delay curve was generated by aligning the peak response with each scan’s raw data after the normalization and smoothing calculations and the first scanline’s raw data. This process is the 1st round of APR estimation.

After this, the 1st round-estimated delay curve was re-smoothed using the Savitzky–Golay polynomial filter with a window size of 20 (20 pixels for 1.6 mm in the axial direction) [49,50]. The 2nd round-estimated delay curve and the corresponding delayed simulated image were obtained. The estimated accuracy of the APR was evaluated by calculating the total errors of each scanline between the ground truth delay and 2nd estimated delay and corresponding mean error. The total delay errors between the ground truth delays and 2nd round-estimated delays were calculated by summing all 128 scanlines’ quantity variances in the delay samples. The corresponding mean errors were obtained by calculating the average value of the total delay errors.

### 2.3. In Vitro Ultrasound Images Experiments

A prototype of a flexible array transducer (made by Hitachi and Japan Probe) was used and connected to the Vantage 128 system (Verasonics Inc., Kirkland, WA, USA). The parameter details of this transducer are shown in Figure 3. One element was used for transmitting and one hundred and twenty-eight elements were used for receiving. The overall workflow for these experimental ultrasound images is illustrated in Figure 4.

First, for simulating the assistant structure within markers, we created a home-made transparent phantom with a four-needle configuration evenly distributed using a gelatin and agar mixture (Figure 5). The height of this assistant structure was approximately 17 mm. The four needles were distributed manually, ensuring that they were evenly spaced without exact measurements in the middle depth of the assistant structure. We organized two types of experiments and collected corresponding RF data by placing this home-made assistant structure on top of different phantoms.

As Figure 5A shows, we first collected RF channel data, where the flexible array transducer was configured in a flat shape on a small parts ultrasound CIRS phantom (Model 050, Computerized Imaging Reference Sys213 terms Inc., Norfolk, VA, USA). This configuration made the flexible array transducer comparable to a linear array transducer. Then, the collected RF channel data from the linearly configured flexible array transducer were artificially curved with varying transducer geometry assumptions to mimic the different configurations of a flexible array probe. Specifically, the initial (linear) time delay and the expected (artificially curved) time delay with specific transducer geometry were calculated for each focal point. According to the time differences, the initial linear RF data were delayed, forming the curved RF data that were used in the in vitro ultrasound image experiments with known ground truth delay. This artificially curved initial (linear) RF data method was used and evaluated in our previous work [27]. This method of calculating delays for reconstructing ultrasound images is identical to the conventional DAS beamformer, which computes delays based on the known transducer shape and the precise positions of each element.

Secondly, as Figure 5B,C shows, we placed the flexible array transducer on the top of an ABDFAN Abdominal Ultrasound Phantom (Kyoto Kagaku Co., Kyoto, Japan) with the home-made assistant structure in the sagittal direction for mimicking the arbitrary shapes of a patient’s body. The Flexible Transducer with External Tracking (FLEX) system, previously proposed in our research [52], was employed to acquire the ground truth for the arbitrary shapes of the probe. The FLEX system comprises five specially designed tracking marker bars that are affixed to the surface of the flexible transducer probe, with each bar containing three optical tracking markers. In using an infrared camera (Polaris, NDI, Waterloo, ON, Canada), the arbitrary shape of the flexible transducer was captured and recorded as the ground truth. This information was then employed by the conventional DAS beamformer to calculate the delays, ultimately reconstructing the corresponding B-mode ultrasound images. The feasibility and accuracy of the FLEX system have been thoroughly tested and assessed in our prior research [52].

For the curved experimental ultrasound RF data, the same RF data normalization was performed similarly to that in the simulations. Since experimental ultrasound data included more noise and various additional echo responses when compared to the simulation data, after routine normalization, the experimental ultrasound RF channel data were also divided by the maximum value of the specific range around the marker’s depth, and the axial position was around [9 mm, 24 mm].

Figure 6 illustrates the workflow of the two-round APR delay estimation process for experimental ultrasound RF data. In this setup, each scanline consists of 128 columns of RF data, received by 128 elements. The process begins by extracting the raw data received by the transmitting element from each scanline, which serves as the reference RF data for estimating the transmitted delays.

In the first round of the APR procedure, we align the peak response of each reference RF data within a specific range (from 9 mm to 24 mm), which corresponds to the location of the markers, to estimate the reference delays for each scanline. Following the application of the first round of APR to each scanline, the second round focuses on estimating the delayed ultrasound image and the corresponding second delays. This is achieved by aligning the peak response between the raw data in each scanline again within the range of 9 mm to 24 mm. The second delays are approximated to represent the received delays.

As noted above, experimental ultrasound image data present more complex and diverse responses, along with noise, especially due to high disorder responses originating from the bottom of the interface between the assistant structure and the phantom surface. To mitigate the risk of outliers in the estimated delay, a threshold value of 5 was set for the estimated received delays between two sets of raw data. The total estimated delays for each column are calculated by summing the referenced delays obtained from the first round of APR with the individual delays from the second round of APR. After applying the APR delays, the summation calculation across the element dimension and standard post-processing steps, including envelope detection and log compression, are performed to reconstruct the final B-mode ultrasound images.

### 2.4. Evaluation Metrics

In this study, we used the lateral FWHM of the 5th point (located in the 50 mm within phantom), as the objective point target (OPT), for assessing defocusing and distortion. In addition, the contrast-to-noise ratio (CNR) was used to evaluate the quality of an image. The CNR can be calculated based on the inner and outer regions defined in the Equation (3):(3)$$CNR={20\log}_{10}\left(\frac{\left|\mu_{out}-\mu_{in}\right|}{\sqrt{\sigma_{out}^{2}+\sigma_{in}^{2}}}\right)$$
where $\mu_{in}$ and $\mu_{out}$ represent the mean signal within and outside the OPT, respectively, and $\sigma_{in}$ and $\sigma_{out}$ represent the standard deviation of signals within and outside the OPT, respectively.

Additionally, we propose an evaluation metric, termed Location Error (LR), to quantify the discrepancy between the ground truth location of the OPT and its position in the corresponding reconstructed ultrasound images obtained using the APR beamformer. This LR metric specifically addresses misalignments in the *z*-axis direction. The formula for calculating the LR is as follows (4):(4)LR=ground truth of OPT location − OPT location in the evaluated ultrasound images (mm)

## 3. Results

Figure 7 shows three example results of in vitro RF data collected based on a small CIRS phantom with an assistant structure. The first example RF data point was collected by a transducer in a flat shape. The other two example results are shown with the artificially curved RF data points with radii in 700 mm and 500 mm, respectively. The reconstructed ultrasound images with two-round APR-estimated delays were compared with the RF data without delays and the ground truth of the reconstructed ultrasound images, respectively. The fifth point target was considered the OPT. Compared with the RF data without delays, the target point was reconstructed and showed around the correct location (67 mm) in reconstructed ultrasound images with APR-estimated delays (the target point locations were 67 mm, 68 mm, and 68 mm based on the RF data collected in a flat shape, artificially curved with radii 500 mm and 700 mm, respectively). But there were certain amounts of distortion and defocusing of this target point observed in the APR-reconstructed ultrasound images compared with the ground truth of reconstructed ultrasound images. The FWHM, CNR and PSNR values of these ultrasound image results are listed in Table 1. The lateral and axial FWHM of the OPT in the ground truth of the ultrasound image were 3.96 mm and 1.52 mm, respectively. The lateral FWHM values of the OPT in the APR-delayed ultrasound images with a flat shape, curved with radii 500 mm and 700 mm were 4.95 mm, 4.96 mm, and 4.95 mm, respectively. The axial FWHM values of the OPT in the APR-delayed ultrasound images with a flat shape, curved with radii 500 mm and 700 mm were 4.08 mm, 5.84 mm, and 5.92 mm, respectively. The CNRs of the OPT in the ground truth of the reconstructed ultrasound image, APR-delayed in a flat shape, artificially curved with radii 500 mm and 700 mm were −13.56 dB, −14 dB, −14.05 dB, and −12.28 dB, respectively.

Figure 8 shows two example results of experimental ultrasound RF data points collected based on the ABDFan phantom with two different arbitrary transducer shapes. In the first example, compared with the ultrasound images without delays shown in the first column in Figure 8, the APR-estimated delays were able to reconstruct ultrasound images with arbitrary shapes. The vein target was shown at the correct depth (around 60 mm). However, compared with the ground truth of the reconstructed ultrasound images, the vein target was reconstructed with distortion to some extent. In the second example, the wrongly delayed estimation from the APR method occurred in the phantom patient‘s left side based on the reconstructed ultrasound images.

## 4. Discussion

According to the results in this study, we can estimate the requirements of an assistant structure with embedded markers that could be used for applying the APR method. The APR method obtained accurate delay estimations with an assistant structure constructed from a homogenous material and embedded with one strong reflection marker with an appropriate size in the center’s shallow depth. This is because of five reasons. (1) As the results show, homogenous background material will not cause unnecessary irrelevant responses that would affect the accuracy of the APR method. This is because the diverse responses reflected from the background material are not reflected as a result of the marker, and thus, they will not be used to estimate the time delays. The stronger and more consistent responses can be better detected using the APR method. This is critical to the performance of the APR method. This also explains how the results of the simulated point target phantom display the ideal simulated ultrasound image with APR-estimated delays compared with a simulated scatter phantom and experimental ultrasound images. (2) The marker embedded within the assistant structure’s material should be located properly in the structure for an optimal APR performance. This is because all the elements within a flexible transducer have to receive responses from the reflective markers but no scatter responses from two neighboring markers. In the Field-II simulations, the response from a marker on one side is not strong enough to estimate the correct delays on the other side of the flexible array probe’s elements, as the flexible probe has a larger FOV (probe width is 128 mm) compared to the conventional rigid array (typically, the linear probe width is 4 mm–6 mm) [53]. Similarly, the wrongly APR-estimated delays in experimental ultrasound images with arbitrary shape are probably because of weak responses reflected on the left side of the assistant structures. (3) As the simulated results show, the accurate delay estimation is generated using the APR if one marker is located in the shallower depths of the assistant structure. In contrast, according to the simulation results, having multiple markers will cause response scatters and interference, thereby reducing the accuracy of the APR method. But in the in vitro ultrasound image experiment results shown in Figure 7 and Figure 8, less markers cannot generate enough strong peak responses for covering the whole FOV. So, there is a compromise between the scatter responses from multiple markers and the strong peak responses. (4) The results based on simulated scatter phantom simulations prove that the APR method estimates delays more accurately when the responses are of higher amplitudes from hyperechoic markers. The responses reflected by hypoechoic markers are not significant enough to be distinguished as a peak response to apply the APR method. (5) The optimal marker size should keep the balance between reflection response and noise scattering, where the response should be maximized, and the noise scattering should be minimized.

There are two primary differences between Field-II simulations and experimental ultrasound images. First, Field-II simulations are designed to solely emulate the assistant structure with point markers, excluding any simulated phantom beneath the assistant structure. This design aligns more closely with clinical application hypotheses and serves to initially evaluate the feasibility of the APR beamformer concept using Field-II simulations. In contrast, subsequent experimental ultrasound images incorporate an actual assistant structure with point markers placed on top of either the CIRS phantom or the ABDFan phantom. This setup demonstrates the accuracy of the APR beamformer in reconstructing ultrasound images under conditions that more closely resemble clinical scenarios.

Based on the results obtained from the simulated assistant structure with point markers, we have demonstrated the primary feasibility of the APR beamformer for ultrasound image reconstruction. However, the simulated assistant structure relies on an ideal anechoic background material, and the point markers reflect perfect “wing”-shaped peak signals. Consequently, the conclusion drawn from Field-II simulations is that the APR beamformer holds promise for reconstructing ultrasound images, provided that peak signals reflected by point markers are received by all elements and no interactions occur between two point markers.

In the in vitro ultrasound imaging experiments, the custom-made assistant structure we created is not anechoic like the simulated one. Consequently, the needle markers were unable to generate pronounced “wing”-shaped peak signals similar to those produced in the simulations that could be received by all elements, leading to no interaction between two point markers as well. Therefore, even though a single point marker within the assistant structure produced the best reconstructed ultrasound images in the Field-II simulations, we opted to insert four needles instead of just one within the home-made assistant structure for the experimental ultrasound images.

Another difference between the Field-II simulations and experimental ultrasound images lies in the fact that experimental RF data contain more noise as well as more complex and diverse signals, which are reflected from the CIRS phantom/ABDFan phantom beneath the assistant structure. As a result, the APR method was applied differently to achieve the most accurate delay estimations. But the fundamental principle of the APR method, as depicted in Figure 1, remains consistent for both simulated and experimental ultrasound images.

The overall results in this study demonstrate that strong received responses and consistent RF channel data are the two major factors that influence the APR estimated delay results. To enhance responses, normalization should be performed as a pre-processing step. In experimental ultrasound images, normalization is not only aimed to enhance the response of the whole scanline but to also enhance the response reflected from the markers. Thus, the embedded marker is needed to properly locate at a specific depth in the assistant material and the maximum response value can be extracted from RF data and can be utilized to perform the normalization. Additionally, with a known marker location, the APR method can be implemented in smaller ranges, around the marker location’s depth, in order to improve the accuracy of estimated delay results.

To improve the consistency of RF data, we applied smoothing functions to the pre-processing procedure. Due to the reflective properties of the assistant structure’s material and hyperechoic marker, noise and scatter responses will be generated, which could negatively impact the results of the APR method. In addition, the consistency of RF data will also vary based on different apodization and aperture growth settings, thereby further decreasing the accuracy of the APR. The smoothing calculations of RF channel data as pre-processing is an effective way to address this problem. The application of different smoothing methods and tuning the associated window size are critical to remove unnecessary noise and scatter responses. For different phantom types, apodization and aperture growth settings, and marker sizes/amplitudes, different smoothing method selections and tunings of associated window sizes are required. However, if the assistant structure is made of a homogenous material, the APR-estimated delay will not depend on the performance of the smoothing method as significantly. This is also the reason why the internal markers of patients’ bodies cannot be considered as a source to generate strong responses for the APR method. The internal markers are located at a random depth and the response from around normal tissue is also ‘random’. Thus, it is not easy to target a specific APR range to enhance the responses of markers, thereby improving the estimation accuracy.

However, the assistant structure with embedded markers and the APR method has their limitations. Such limitations of the proposed assistant structure and APR methods include that the estimated delays still contain errors because of the uncertainty caused by noise and the inconsistencies of reflecting responses. Thus, the estimated delay from the APR method will need further correction to obtain the best reconstructed ultrasound images. More troubling, a reasonable pre-processing workflow will increase the calculation time, which is not favorable for clinical applications. Furthermore, as the results were based on the ABDFan phantom, the wrong delay estimation on the phantom patient’s left side was applied to reconstruct the ultrasound image in the upright corner. This is because the peak response reflected by the markers is weak on the edges of the images. The tradeoff between sufficient peak responses (covering all elements) and the image reverberation caused by the markers is important when designing the assistant structure. Lastly, there is the main disadvantage of using an external assistant structure within markers. The markers within the assistant structure can potentially create image artifacts.

In future work, the enhancement of ultrasound image quality obtained using a flexible transducer probe will necessitate advancements in both hardware and image reconstruction and processing methodologies. With a focus on image reconstruction and processing, (1) given that our flexible array transducer is still a prototype lacking advanced post-image processing, the ultrasound images exhibit ‘scatters’ at the interface between the assistant structure and the phantom. Developing sophisticated post-image processing methods is essential to improving image quality by reducing the reverberation and scatter caused by the markers and interface. (2) Additionally, the delays estimated using the APR method resemble those of a fixed focusing beamforming rather than a dynamic delay. In the in vitro ultrasound image experiments, the first and second rounds of APR estimation can be viewed as transmitting and receiving delay estimations, respectively. However, these estimations are based on peak responses from reflected markers, similar to fixed focusing beamforming. This approach limits the reconstructed ultrasound images to a high resolution and accuracy at a specific depth only. Integrating dynamic focusing, which is utilized in conventional beamformers like DAS and synthetic aperture beamforming, could significantly enhance the image axial resolution and quality across the entire image depth. These conventional beamformers calculate delays based on known element positions, but our APR method currently does not support the direct incorporation of dynamic focusing. A possible solution could involve using the APR principle to inversely estimate the positions of the transducer elements, and then reconstruct the ultrasound images using dynamic focusing in a conventional manner. (3) The most accurate delay estimation using the APR method was achieved with one active element in the emitting transducer and 128 active elements in the receiving transducer. However, this configuration, without apodization and aperture growth, compromises the lateral resolution of the ultrasound image, as the ratio between the focal depth and the size of the receiving aperture is not maintained. Therefore, effectively applying the APR method requires balancing between accurate time delay estimations and the resolution limitations of the ultrasound images.

In conclusion, our method was tested using Field-II-simulated data and two experimental ultrasound image datasets based on the CIRS and ABDFan phantoms. To fully assess the feasibility and efficiency of this approach, additional experiments and tests are necessary. The ultimate goal of employing this flexible ultrasound transducer technology is to monitor moving targets during radiation therapy, allowing for the precise delivery of radiation doses to tumor targets while minimizing exposure to surrounding healthy tissue. In this study, we evaluated the use of the APR beamformer with the flexible transducer to reconstruct ultrasound images by focusing on stationary targets. However, we did not validate the method using a phantom test with moving targets, which marks a limitation of our current work. Our findings suggest the potential and promise of employing this technique for the real-time tracking of moving targets, despite this limitation.

Additionally, it is important to note that our flexible array transducer prototype was designed as a two-dimensional array, not a three-dimensional one. Therefore, we tested the potential of this technology for tracking one-dimensional movements, specifically in the superior–inferior direction, which previous research has identified as the primary direction of respiratory-induced tumor motion [25]. While the current study does not offer a comprehensive solution for monitoring respiratory movements in 3D using three-dimensional ultrasound images, it demonstrates sufficient potential, especially when combined with gating technology, to improve the accuracy of radiation delivery and guide treatment decisions in the future.

## 5. Conclusions

The assistant structure with embedded markers and the APR method proposed in this study present a novel alternative for delay estimation, aiming to replace the conventional DAS beamformer for reconstructing ultrasound images from a flexible probe. The results from both simulated and experimental ultrasound image RF data have shown that the proposed assistant structure and APR method are capable of constructing accurate delay curves. This opens up the possibility of applying this method to flexible array probes during radiation therapy, specifically for the real-time tracking of pancreatic tumor targets. In the future, further examinations are needed to test this proposed APR method on real-time moving targets.

## Figures and Tables

**Figure 2 cancers-16-01244-f002:**
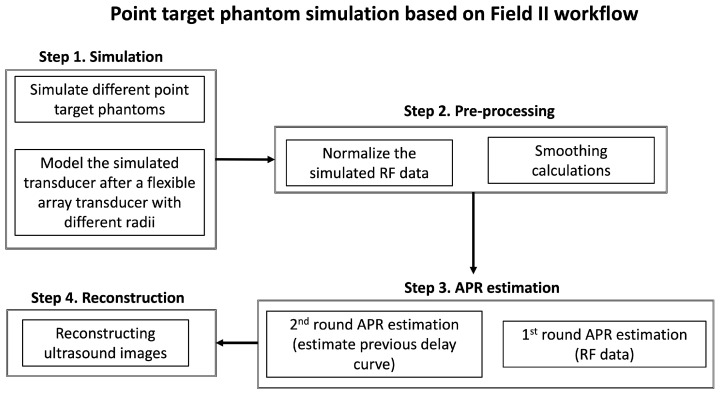
The overall workflow of the Field-II simulation. The four steps included in this workflow are as follows: Step 1. Simulating various point target phantoms and different flexible transducer shapes. Step 2. Pre-processing, which includes normalization and applying smoothing filters. Step 3. Two rounds of APR estimation. Step 4. Reconstructing the ultrasound images based on the APR estimation.

**Figure 3 cancers-16-01244-f003:**
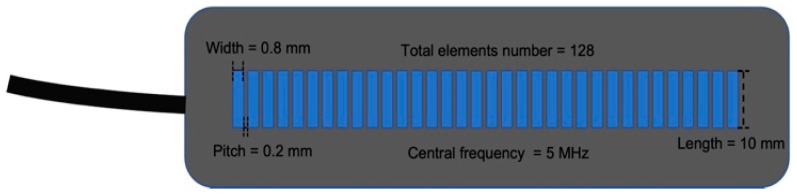
Transducer parameters of flexible probe. The parameters include the width and length of each element, pitch, number of elements, and central frequency.

**Figure 4 cancers-16-01244-f004:**
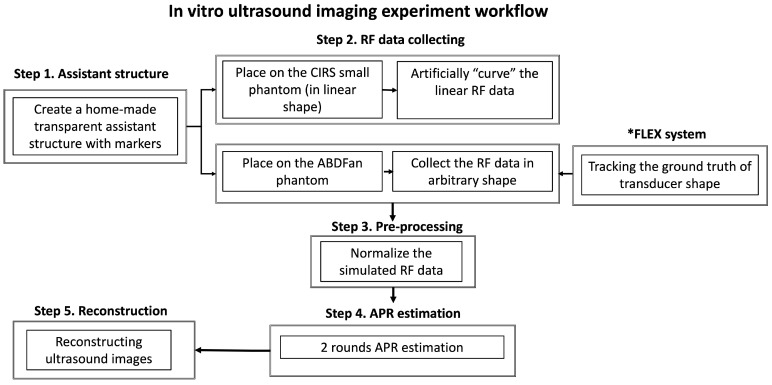
The overall workflow of the experimental ultrasound imaging. Step 1: creating a home-made transparent assistant structure with markers. Step 2: RF data are collected based on the CIRS phantom and ABDFan phantom, separately. Step 3: Pre-processing, which includes normalization calculation. Step 4: Two rounds of APR estimation. Step 5: Ultrasound images are reconstructed. * represents the FLEX system developed in our previous work [51].

**Figure 5 cancers-16-01244-f005:**
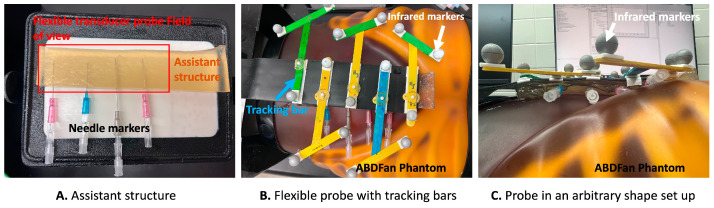
In vitro ultrasound imaging experiments set up. (**A**) shows the home-built assistant structure with four needle markers. (**B**) shows the setup of using five infrared camera-tracking bars that we designed. (**C**) shows the flexible probe placed with the assistant structure on top of the ABDFan phantom for collecting raw data with arbitrary shapes.

**Figure 6 cancers-16-01244-f006:**
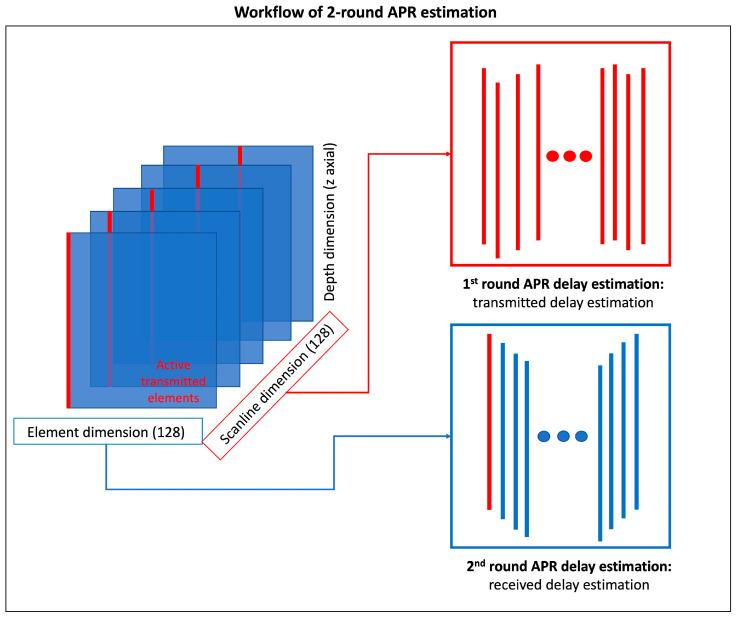
Workflow of two-round APR estimation in in vitro ultrasound image experiments. The 1st round of APR estimation was implemented in the sub-group for estimating transmitted delays (red frames). The 2nd round of APR estimation aimed to calculate the received delay in each scanline (blue frame).

**Figure 7 cancers-16-01244-f007:**
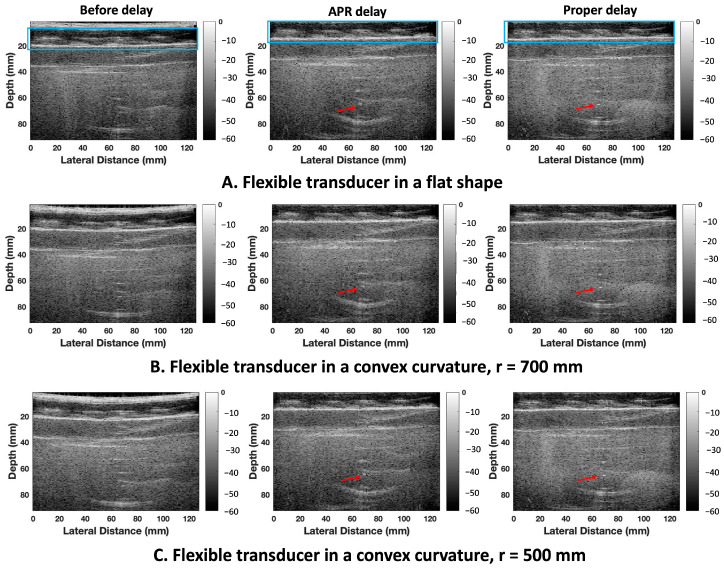
Comparison of the in vitro raw ultrasound image data without delay, a reconstructed ultrasound image with two rounds of the APR method, and the ground truth of a reconstructed ultrasound image. Figure 7A–C shows three example ultrasound image results collected in a flat shape and convex curvature shape with radii of 700 mm and 500 mm, respectively. All the ultrasound images are shown in a dynamic range from −60 to 0 dB. The assistant structure is contoured with a blue frame in the example result with a flat shape. The evaluated target point, the seventh point target located in the 70 mm, is pointed at by the red arrow (see the Appendix A).

**Figure 8 cancers-16-01244-f008:**
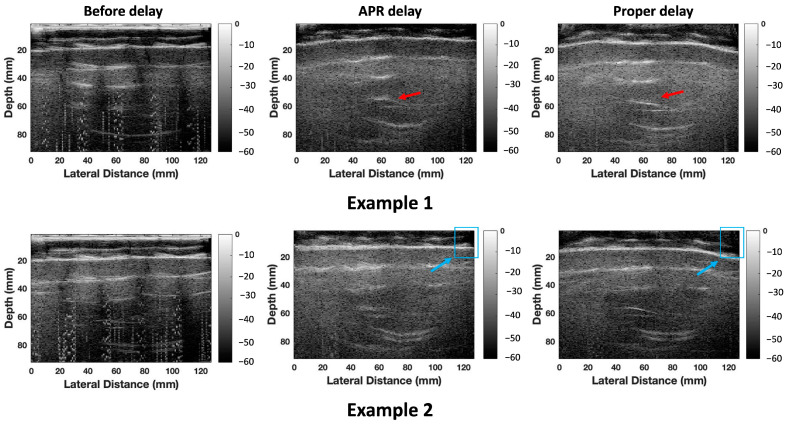
A comparison of reconstructed ultrasound images without delays, with APR-estimated delay and the corresponding ground truth of the reconstructed ultrasound images. Two example results are shown with the raw data collected with a flexible probe in two arbitrary shapes based on the ABDFan phantom. The red arrows point to a vein target within the ABDFan phantom for evaluating the image quality. The blue arrows and blue frame contour the wrongly delayed part with APR-estimated delays compared with the ground truth of the reconstructed ultrasound images. All the ultrasound images are shown in a dynamic range from −60 to 0 dB.

**Table 1 cancers-16-01244-t001:** The evaluation of the experimental ultrasound images.

Geometry of Flexible Array	Lateral FWHM [mm]		Axial FWHM [mm]		CNR [dB]		PSNR [dB]	Depth in the *z*-Direction	
	APR	Ground Truth	APR	Ground Truth	APR	Ground Truth	PSNR for APR Images Compared to Ground Truth	APR	Ground Truth
Flat Shape	4.95	3.96	4.08	1.52	−14	−13.56	−19.05	67	67
Convex with a Radius of 500 mm	4.96	3.96	5.84	1.52	−14.05	−13.56	−19.14	68	67
Convex with a Radius of 700 mm	4.95	3.96	5.92	1.52	−12.28	−13.56	18.94	68	67

## Data Availability

The data supporting the findings of this study are available from the corresponding author upon reasonable request. This study involves simulation data that, due to its nature, is not publicly archived but can be provided to interested researchers subject to an evaluation of the request’s validity and the potential for collaborative research.

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
