# Peer review of "Enhancing Image-Guided Radiation Therapy for Pancreatic Cancer: Utilizing Aligned Peak Response Beamforming in Flexible Array Transducers"

_cancers, 2024, doi:10.3390/cancers16071244_

Round 1
Reviewer 1 Report
Comments and Suggestions for Authors
The topic is important and worth consideration, the manuscript is enought fixalbe to answer the aim of the study.
many thanks for the honor to be invited to review the article "Enhancing Image-Guided Radiation Therapy for Pancreatic Cancer: Utilizing Align-Peak-Response Beamforming in Flexible Array Transducers ". I have read the manuscript with interest and my suggestion is to accept it with a minor revision.
The challenge of target motion and surrounding organs in radiotherapy treatment of pancreatic cancer is extremely relevant. The authors aim to evaluate a novel and alternative method of ultrasound imaging, align peak response (APR), to construct a flexible ultrasound array for tracking pancreatic tumor targets in real-time for radiotherapy. The topic is important and worth consideration. The results are enough fixable to answer to the endpoints. Some minor revision is required with moderate editing of English language.
For more specific comments regarding the technique, a review by a medical physicist expert in image-guided is suggested.
Comments on the Quality of English LanguageModerate editing of English linguage is required.
Author Response
Thank you for your valuable feedback on our manuscript. We appreciate your comments regarding the need for moderate editing of the English language to enhance the clarity and readability of our work. In response to your suggestion, we have revised our manuscript and marked all changes in red in the revised manuscript. This is to facilitate your review process by making it easier to identify the modifications made in response to your comments and suggestions.

Reviewer 2 Report
Comments and Suggestions for Authors
The manuscript by Feng et al. describes about a method to improve image-guided radiation treatment for pancreatic cancer.
Some suggestions to improve the manuscript are:
1. The legend for Figures 2, 3 and 5 should be significantly elaborated.
2. The images in Figure 7 are too blurred. Kindly improve/enhance the resolution if possible.
3. The conclusion section should include some future directions.
Author Response
We greatly appreciate the time and effort the reviewer has dedicated to evaluating our manuscript titled "Enhancing Image-Guided Radiation Treatment for Pancreatic Cancer." Your insightful comments have been invaluable in guiding our revisions. Below are our responses to each of the suggestions:
- Elaboration of Legends for Figures 2, 3, and 5
Response: We acknowledge and appreciate the reviewer's suggestion to elaborate on the legends for Figures 2, 3, and 5. In response, we have significantly expanded each legend to provide a more detailed description of the depicted results and workflow. These enhancements aim to ensure that the figures are comprehensible and informative to readers without the need to refer back to the main text. The updated legends for Figures 2, 3, and 5 are highlighted in red, and can be found on lines 197 - 200, 235 - 236, and 271 - 275, respectively.
- Improvement of Image Resolution in Figure 7
Response: We are grateful for the reviewer's identification of the issue with the image quality in Figure 7. The reduction in image clarity, resulting from the necessary resizing to meet the manuscript's formatting requirements within the Word document, inadvertently compromised the figure's resolution. To rectify this, we have opted to submit Figure 7 as a separate high-resolution file alongside our revised manuscript. (please see the attachment) This approach ensures that the figure retains its original quality and detail, facilitating clearer visualization and interpretation of the results. We believe this solution effectively addresses the concern raised and upholds the integrity and readability of our visual data.
- Inclusion of Future Directions in the Conclusion Section
Response: The suggestion to include future directions in our conclusion section has been well-received. We have revised the conclusion to incorporate potential future research directions emerging from our study. The updated conclusion is highlighted in red (lines 547 -- 555).

Reviewer 3 Report
Comments and Suggestions for Authors
The manuscript by Ziwei Feng et al. aims at improving the accuracy of ultrasound-guided RT through a novel method of reconstructing ultrasound images. The study, in my opinion, is accurately conducted and the data are presented clearly and comprehensively. The conclusions, although preliminary, are supported by an extensive description of the method. The bibliography is exhaustive. I believe the paper can be published in the present form.
Author Response
Thank you for your positive feedback and support for our manuscript. We're grateful for your endorsement for publication.
Reviewer 4 Report
Comments and Suggestions for Authors
I really offer my congratulations to the authors because they treat an interesting and new topic that has a relevance to treat pancreatic cancer using radiotherapy. I have no suggestions to improve the article that appears already set for publication
Author Response
Thank you for your congratulations and recognition. We appreciate your support for our manuscript's publication.